# Effect of Ball Milling Time on the Microstructure and Properties of High-Silicon–Aluminum Composite

**DOI:** 10.3390/ma16175763

**Published:** 2023-08-23

**Authors:** Zhaoyang Kong, Zhipeng Wang, Bin Chen, Yingmin Li, Runxia Li

**Affiliations:** 1School of Material Science and Engineering, Shenyang University of Technology, Shenyang 110870, China; 2023123@dgut.edu.cn (Z.K.); liym@sut.edu.cn (Y.L.); 2School of Materials Science and Engineering, Dongguan University of Technology, Dongguan 523808, China; wangzp1205@hnu.edu.cn (Z.W.); chenbin_ustc@163.com (B.C.)

**Keywords:** electronic packaging materials, thermal conductivity, spark plasma sintering, thermal expansion coefficient

## Abstract

The duration of ball milling greatly influences the characteristics of high-silicon–aluminum composite during the ball milling process. This study examines how the microstructure, thermal conductivity, and hardness of a high-silicon–aluminum composite are affected by different ball milling times. We exposed the powder to various durations of ball milling and employed different pellet ratios. Following this treatment, the powder underwent consolidation via discharge plasma sintering. Our findings show that with a pellet ratio of 10:1 and a milling duration of 8 h, the powder particles were refined, resulting in a more uniform and dense material composition. This refined material boasted a thermal conductivity of 111.6 W/m·K, a Brinell hardness of 136.8 HBW, and a density of 2.304 g/cm^3^. This method facilitates the creation of a uniform composite powder composition. It encourages the development of a fine-grain structure, which enables the production of particle-reinforced composites with superior properties.

## 1. Introduction

High-silicon–aluminum composite, as an electronic packaging material, has many advantages, such as a relatively small density, high thermal conductivity, low coefficient of thermal expansion, good mechanical properties, good weldability, easy precision machining, non-toxicity, etc. Hence, high-silicon–aluminum composite materials for electronic packaging fulfill the requirements of modern electronic packaging development [1,2,3,4,5]. Modifying the composite composition ratio can also enhance certain physical properties while maintaining cost-effectiveness. These high-silicon–aluminum composites hold great potential for diverse applications in aviation, aerospace, and national defense [6,7]. The Sumitomo Electric Corporation in Japan utilized powder metallurgy hot extrusion to create an Al-40 wt.% Si composite with a CTE of 13 × 10^−6^·K^−1^, TC of 126 W/(m·K), and density of 2.53 g/cm^3^ [8]. Wang et al. [9] successfully prepared an Al-70 wt.% Si composite through extrusion casting and hot-pressing sintering, resulting in a three-dimensional continuous network distribution of a reinforcing phase with a relatively dense microstructure and some defects. Nonetheless, the primary obstacle arises from the material preparation process, limiting their broader application. For instance, high-silicon–aluminum composites comprise sizable and brittle primary silicon particles. These particles induce local stress concentration because of their dimensions and fragility. Consequently, crack initiation, propagation, and even fracture occur, significantly reducing the composite’s properties [10]. Therefore, it is crucial to improve the preparation method, aiming to refine these large and brittle silicon particles and enhance the mechanical properties of high-silicon–aluminum matrix composites [11,12,13].

Ball grinding is considered as an effective method that transforms large particle powder into fine particles via high-speed rotation or vibration and generates a strong impact and grinding between the grinding ball and raw materials [14,15]. The vigorous collision and friction result in a pristine atomized surface on the powder particles [16,17]. After ball milling, the surfaces of these newly created particles have elevated surface energy. Under the sustained pressure from ball milling, the fragmented powder pieces interlink and bind, leading to the formation of lamellar composite particles or secondary particles. These are interconnected by specific atomic binding forces within the lamellar tissue. This process of particle fragment aggregation is commonly termed as cold welding. After this welding process, the powder particles continue to be influenced by the previously mentioned forces, experiencing ongoing transformations. The cyclical processes of welding and breaking lead to continuous densification of the powder particle structure and a refinement in grain size [18,19,20,21]. Various intense forces introduced during ball milling infuse a significant amount of strain into the powder particles, accumulating a host of defects like vacancies and dislocations between the layered structures, which results in high lattice distortion energy and surface energy within the powder particles, thereby augmenting their surface activity and internal energy. The atomic diffusion and reactions between and within the powders are more likely to occur on the newly formed microstructure surface during the ball milling process, producing composite powder particles [22,23,24,25].

Ashutosh Sharma et al. [26] conducted ball milling of pure Al powder and pure SiO_2_ powder at 300 r/min, using a ball material ratio of 10:1 and ball milling time of 20 h. Subsequently, they successfully prepared a composite material with an Al-12 wt.% Si composition ratio through low-temperature sintering. In another study by Hansung Lee et al. [27], Fe, Co, Ni, Al, and Si powders were subjected to ball milling with a pellet ratio of 10:1 and a ball milling time of 48 h. This enabled the successful preparation of FeCoNiAlSix (x = 0, 0.2, 0.4, 0.6, 0.8) series high entropy alloys using spark plasma sintering. Whereas there have been numerous studies on ball milling, research on composites with high Si content remains limited. To address this gap, the present experiment focuses on Al-60wt.%Si as the research subject to explore the effects of ball milling on high-silicon–aluminum composites. Different ball milling times (2 h, 4 h, 6 h, 8 h, 10 h, and 12 h) were employed as pretreatment for the powder, aiming to determine the optimal powder milling time. Following this, the preprocessed powder was subjected to discharge plasma sintering to fabricate high-silicon–aluminum composites. This study aims to examine the effects of ball milling duration and pellet ratio on the oxygen content, the morphological attributes of the composite powder particle size, and the microstructure of the produced material.

## 2. Materials and Methods

An Al-60 wt.% Si composite material (Tenghui Metal Material Co., Ltd., Xingtai, Hebei Province, China) was fabricated by employing pure Al powder and pure Si powder. Table 1 displays the composition and particle size of the powders utilized in this study. Figure 1 illustrates the morphologies of the initial aluminum powder and the silicon powder.

The pure Al powder and pure Si powder were proportionally mixed. The grinding process employed zirconia balls as the grinding medium, while the grinding tank was constructed using polytetrafluoroethylene and had a capacity of 1L. The ball mill used was a XQXM-2 planetary ball mill. A consistent ratio of 10:1 for the ball material was upheld during the ball milling process, and a grinding speed of 200 r/min was sustained. The grinding periods were differentiated, with durations set at 2 h, 4 h, 6 h, 8 h, 10 h, and 12 h, respectively. A loading coefficient of 0.4 was established. Analysis of the powder’s particle size was conducted utilizing a HITACHI S-3400N scanning electron microscope (SEM) combined with an energy-dispersive spectrometer (EDS) (Suzhou Sainz Instrument Co., Ltd., Suzhou, Jiangsu Province, China). The particle size measurement of the powder was conducted using the Malvern laser particle size analyzer MS2000 (Suzhou Sato Precision Instrument Co., Ltd., Suzhou, Jiangsu Province, China). For oxygen content analysis, a nitrogen-oxygen analyzer with a power of 5000 W, analysis time of 30 S, and gas flow rate of 300 mL/min was utilized (Beijing Xibi Instrument Co., Ltd., Beijing, China).

Following ball grinding, the powder was loaded into a graphite mold and subsequently placed on the sintering platform within a vacuum chamber for spark plasma sintering. The sintering process encompassed two stages of temperature rise. The initial stage involved a temperature increase at a rate of 50 °C/min until reaching 520 °C. In the second stage, the temperature gradually increased at a rate of 5 °C/min until reaching 550 °C during the final 30 °C increment. A pressure of 30 MPa was applied throughout the heating process, and the sintering temperature was maintained for 1 h prior to cooling (Figure 2).

A tissue block of 10 mm × 10 mm × 10 mm was obtained from the sintered block and subjected to polishing. The sample was analyzed using the HITACHI S-3400N SEM equipped with EDS equipment. The Tecnai G2 F20 high-resolution transmission electron microscope (HRTEM) (Shenzhen vector scientific Instrument Co., Ltd., Shenzhen, Guangdong Province, China) and JEOL 2000FX Transmission electron microscopy (TEM) (Shenzhen blue star Yu electronic Technology Co., Ltd., Shenzhen, Guangdong Province, China) were employed to further examine the Al-Si interface. Thin-section TEM samples were fashioned utilizing a focused ion beam (FIB) apparatus. Phase analysis of the samples was carried out with an X’Pert type X-ray diffractometer (XRD) (Shanghai Sibaiji Instrument System Co., Ltd., Shanghai, China). The hardness of the material was determined using an HBS-3000B digital display Brinell hardness tester (Shanghai Yi Longitudinal precision Instrument Co., Ltd., Shanghai, China). Density measurements were conducted using the drainage method, and thermal conductivity was determined using a DXF-200 thermal conductivity meter (sample size: Φ12.72 mm) (Shenzhen sunway technology Co., Ltd., Shenzhen, Guangdong Province, China). The material density was measured using an electronic balance with an accuracy of 0.0001 g. Furthermore, the thermal expansion coefficient (TC) of the material was assessed using a Unitherna Dilatometer System Series 1000 thermal expansion tester (Beijing SMIC Yunke Instrument Co., Ltd., Beijing, China). The TC test involved heating the Φ5 × 20 mm samples from room temperature (RT) to 150 °C at a heating rate of 2 °C/min.

## 3. Results and Discussion

### 3.1. Effect of Milling Time on Particle Size, Morphology, and Oxygen Content of the Powder

Figure 3 and Figure 4 illustrate the distribution of the average particle size and the morphology of the milled powder over varying durations, respectively. The illustrations reveal a significant reduction in the particle size of the blended powder following 2 h of ball milling, leading to an average particle size close to 16.8 µm. Although the milled particles exhibited an uneven shape, a few nearly spherical particles remained visible. In addition, the beginning stages of particle agglomeration were evident. Upon extending the milling duration to 4 and 6 h ball milling, the particle size of the powder mixture decreased compared with that of the powder after 2 h ball milling, but the powder was still slightly agglomerated. When the milling time reached 8 h, it could be seen that the agglomeration phenomenon was reduced, and the size of mixed powder particles was further reduced. After continuous ball milling for 10 h, the particle size experienced only a slight change. However, some mild agglomeration of the powder still remained. As the milling process continued for 8 h, the agglomeration phenomenon appeared to be mitigated, and the size of the mixed powder particles reduced further. However, after 10 h of sustained ball milling, there was only a slight variation in the particle size. Nonetheless, an extended milling period led to enhanced agglomeration, causing a marked rise in the median particle size.

We undertook an analysis of individual particles in the powder after varying durations of ball milling. Figure 5 presents an expanded SEM image and the associated EDS distribution diagram for selected particles within the powder, milled for different lengths of time. Si is denoted in red in the EDS distribution, while Al is represented in green. The figure reveals that after 2 h of ball milling, Si particles were found in the vicinity of Al particles with minimal contact. After 4–6 h of ball milling, Si started adhering to the Al matrix particles. As the milling time extended, Si progressively embedded itself within the Al matrix particles. Considering the superior hardness of Si particles compared to Al particles, extended ball milling caused the harder silicon particles to slowly embed into the softer aluminum matrix particles, a result of the persistent collisions among the ball, silicon, and aluminum particles. The fracture mechanism dominated the initial phase of ball milling. Under the influence of ball milling, Al and Si particles underwent deformation and fracture, significantly reducing the average size of mixed powder particles. In the subsequent phase, cold welding emerged as the primary factor, leading to slight agglomeration of the powder mixture. By the third stage, an equilibrium between crushing and agglomeration was achieved, and the mean particle size of the powder ceased to undergo significant changes. The cyclical process of breaking and cold welding promoted the uniform dispersal of hard silicon particles within the aluminum matrix, producing consistent silicon particle-reinforced aluminum matrix composites. However, prolonging the milling time beyond a certain threshold upsets this equilibrium. Factors such as the temperature during milling can increase the powder particle size and result in powder agglomeration.

XRD examinations were performed on powder samples subjected to milling durations of 2, 4, 6, 8, 10, and 12 h, with the resulting XRD patterns depicted in Figure 6. Apart from the Al and Si diffraction peaks, the XRD patterns of each milling duration also featured diffraction signals from Al_2_O_3_ and SiO_2_. As the milling time was extended, the intensity of the diffraction peaks for Al and Si was noticeably reduced, and these peaks became wider. This is attributed to the grain size refinement and significant lattice distortion caused by the milling process. According to the Al-Si phase diagram, Al can dissolve Si. It is evident that with the increase in milling time, the intensity of the Al (111) reflection peak diminished, the peak broadened, and the peak shifted to the right (Figure 7) [28].

Figure 8 depicts the correlation between oxygen content and varying durations of post-ball milling in the powder. For comparative purposes, the figure also incorporates the oxygen content of air-oxidized powder over corresponding durations. Under a constant pellet ratio condition, the oxygen content in the powder increased with the duration of ball milling, showing a nearly linear relationship with the milling duration. This can be primarily attributed to four factors: (1) In the ball milling process, the powder is continually crushed, thereby exposing fresh surfaces to oxidation. (2) As the milling time increases, the powder’s particle size diminishes, thereby increasing the specific surface area and surface energy. This boosts the contact interface with atmospheric oxygen, heightening the likelihood of a reaction between the powder and oxygen. (3) Prolonged milling time increases the powder’s temperature, inducing a reaction between the powder and oxygen, and then accelerating the powder’s oxidation process. (4) An extension in milling time results in an increase in the internal defects of powder crystals, which facilitates the diffusion of oxygen within the powder. Furthermore, the findings indicate that the oxidation effect of the powder during ball milling was considerably higher compared to air-oxidized powder at the same oxidation time. The powder displayed a markedly increased oxygen content, largely due to the surface oxidation that transpired during the ball milling procedure. This led to the creation of Al_2_O_3_ or SiO_2_ oxide layers. During the refinement process of ball milling, the powder experienced the continuous disruption of the hard and brittle oxide film, thereby forming an unoxidized fresh surface and enabling the oxidation process to proceed.

### 3.2. Effect of Milling Time on Microstructure and Properties of Materials

#### 3.2.1. Effect of Milling Time on Microstructure

Figure 9 depicts the microstructure morphology after the sintering of Al-60wt.%Si powder at different milling times. It is evident that shorter milling times resulted in the presence of numerous voids at the interface between the silicon and aluminum phases. This occurrence can be attributed to several factors. Initially, the short duration of ball milling for the aluminum and silicon powders results in inconsistent powder distribution and clumping. This, in turn, reduces the wetting capability of liquid aluminum to silicon during sintering, obstructing efficient void filling and leading to imperfections in the sintered product. However, as the milling duration extends to 8 h, the uniformity of powder mixing improves considerably. The milling process breaks down the larger silicon particles into finer angular ones and somewhat refines the aluminum particles. During the sintering process, improving wetting ability promotes a uniform and continuous distribution of the aluminum matrix, leading to reduced internal defects. As ball milling time continues to increase, the presence of voids gradually intensifies. This is due to the uneven distribution of the mixed powder over time, leading to powder agglomeration. The most severe agglomeration was observed at a milling time of 12 h; the holes were gradually increasing because the distribution of mixed powder was uneven with the extension of time. The powder agglomeration occurred, and the agglomeration reached its maximum at 12 h.

Figure 10 displays the SEM image illustrating the material under varying ball milling durations. The size of the silicon phase in the structure exhibited a decreasing trend with the extension of milling time. However, when the milling time reached 8 h, the reduction in the silicon-phase size became less apparent, showing a slight upward trend. Simultaneously, the number of defects in the structure decreased initially and then increased with the milling time. The lowest number of defects in the material structure was observed at 8 h of milling time. The observed pattern aligned with the metallographic microstructure, as previously discussed. To delve deeper, we undertook measurements of the average size of the silicon phase in the microstructure at varying time durations, as shown in Figure 11. Interestingly, at a ball milling duration of 8 h, the material exhibited a somewhat shorter yet larger average size of the silicon phase, with the smallest average size measuring 13.8 µm.

Figure 12 presents the SEM image of the Al-Si interface after 8 h of ball milling. The microstructure of Al and Si, as well as the Al-Si interface, were clearly visible. The interface between Al and Si exhibited a tightly combined, smooth, and flat lattice without any cracks. It can be observed that after 8 h of ball milling, the size of the Al and Si grains decreased, and no Al-Si metal compounds were formed at the interface. However, there was a certain degree of diffusion, indicating that a small amount of Si may have dissolved into Al, and the two phases interpenetrated each other. As a result, a well-bonded Al-Si interface was formed. Subsequently, HRTEM analysis was conducted to examine the interface further. The left half of Figure 13 displays the HRTEM diagram of the sintered powder after 8 h of ball milling. Clear distinctions in lattice orientations can be observed. The interfaces between Al and Si became distinguishable upon applying Fourier transforms to the lattice regions with different orientations. The combination of Al and Si lattice structures at the Al-Si interface indicates the formation of a well-integrated Al-Si interface after prolonged ball milling, suggesting that ball milling does not result in the formation of Al-Si metal compounds at the Al-Si interface; however, there may be some degree of diffusion, leading to a small amount of Si dissolving into Al. The diamond structure of Si particles significantly differs from the aluminum matrix in terms of performance and structure, making interface bonding challenging. Nevertheless, extending ball milling promotes a tight interface bonding. The interface between the matrix and reinforcement plays a crucial role in metal matrix composites. The overall performance of composite material largely depends on the interface bonding quality. Effective interface bonding between the reinforcement and matrix significantly enhances the material’s performance.

#### 3.2.2. Effect of the Milling Time on Properties of Materials

Figure 14a illustrates the thermal conductivity of the Al-Si composite, produced via powder milling across varying time durations. The data indicate that thermal conductivity rose with the ball milling duration, keeping the pellet material ratio consistent. Notably, when the ball milling time reached 8 h, the material achieved a maximum thermal conductivity of 111.6 W/m·K. However, continuous extension of the ball milling time resulted in a decrease in the material’s thermal conductivity to 99.2 W/m·K. This phenomenon can be attributed to two factors. First, high-energy ball milling leads to increased composite oxidation, resulting in the formation of oxide compounds with low thermal conductivity. After undergoing oxidation through hot extrusion, the powder essentially forms a distribution of reinforced particles within the composite material. Second, high-energy ball milling refines the composite powder and introduces significant internal energy, promoting phase flow and rearrangement processes. This improved bonding between composite particles reduces the presence of pores and defects within the material. However, these pores exhibit low thermal conductivity and cause severe phonon scattering, negatively impacting the overall thermal conductivity of the material.

Overall, the limited oxide content in the composite powder and the high-energy ball milling process facilitated solid diffusion and reactions within the material, which enhanced the binding state of phase interfaces, leading to improved uniformity and density. In addition, it effectively refined the second-phase particles in the material, thereby enhancing the connectivity between the matrix and contributing to the overall improvement in thermal conductivity.

In addition to thermal conductivity, the thermal expansion coefficient is another crucial parameter for electronic packaging materials. The thermal expansion coefficient represents an inherent thermal property of the material and plays a key role in determining its compatibility with internal electronic components. Figure 14b illustrates the correlation between the thermal expansion coefficient and varying durations of time. The results indicate an initial increase followed by a decrease in the thermal expansion coefficient as the ball milling time extended. Notably, the highest thermal expansion coefficient was observed when the ball milling time reached 8 h.

The thermal expansion coefficient of a material is influenced by various factors. A higher concentration of reinforcing phases, finer reinforcing particles, and their even distribution within the material induces greater resistance to the thermal expansion of the matrix. Consequently, the thermal expansion coefficient of the material decreases due to an increase in material density and a reduction in residual voids. This is because the non-expansion of pores upon heating can be considered as rigid third phases with zero expansion. Analyzing the previous metallographic structure and density analysis of the material, the material becomes denser and more uniform after ball milling, which adversely affects its thermal expansion performance. Moreover, the presence of oxidation particles also affects the thermal expansion behavior. Both Al_2_O_3_ and SiO_2_ display limited thermal expansion. When they are evenly distributed throughout the material, they restrict the matrix’s thermal expansion, thereby decreasing the thermal expansion coefficient.

The material density is another significant parameter to consider. Figure 15a displays the density and relative density of the Al-60wt.%Si material at different milling times. We note that the density and relative density of the high-silicon–aluminum composite first rose and subsequently declined during the ball milling process. At a milling duration of 8 h, the material displayed a high density of 2.304 g/cm^3^ and a relative density of 93.28%. This is primarily attributed to two key factors. First, as the ball milling time increases, the particle size of the powder becomes smaller. During the high-energy ball milling process, repeated cold welding and crushing refine the internal structure of composite particles. Simultaneously, solid-phase reactions and diffusion strengthen the bonding force between the aluminum matrix and the reinforced particles. However, as the ball milling time continues to increase, it disrupts the equilibrium state, causing the particle size to deviate from its optimal range. This reduction in powder uniformity adversely affects the density and relative density of the material. Second, during the ball milling process, the degree of oxidation increases, resulting in a greater amount of finely dispersed oxides. These oxides can fill the internal pores of the material and reduce its porosity, thereby improving the material’s air tightness. However, considering the overall perspective, the oxide content remains relatively low, resulting in a minimal impact.

Hardness is a crucial characteristic sought in modern electronic packaging materials. Figure 15b presents the Brinell hardness of the high-silicon–aluminum composite at different ball milling times. The results indicate that the Brinell hardness of the high-silicon–aluminum composite gradually increased and then decreased as the ball milling process extended. The highest hardness value, reaching 136.8 HBW, was achieved at a milling time of 8 h. At this condition, the small initial particles in the powder led to the formation of a small grain size after sintering. Based on the Hall–Petch equation, which describes the correlation between the hardness and grain size of polycrystalline substances, reduced grain size typically leads to increased hardness. However, with a milling duration of 12 h, there was a slight drop in hardness, which can be attributed to the rise in the oxide phase presence within the matrix. When the content of the oxide phase exceeds a certain level, it causes local phase agglomeration in the matrix, leading to more defects and reduced material hardness [29].

## 4. Conclusions

This study focused on a high-silicon–aluminum composite with a Si content of 60%. The impact of ball milling time on the final size morphology of the high-silicon–aluminum composite and its microstructure, density, hardness, and thermal physical properties was analyzed. The following conclusions are drawn:(1)The oxygen content in the powder increased with both the increase in pellet ratio and the extension of milling time. The observed rise in oxygen content was considerably greater than that seen in the powder exposed to air oxidation at elevated temperatures over an equivalent period. The milling procedure resulted in a marked decrease in particle size. After milling for 8 h, the powder exhibited the most uniform distribution, with an average particle size of 12.4 µm. Moreover, the prepared materials featured the smallest average size of the silicon phase in their microstructure.(2)At a milling time of 8 h, the Al matrix demonstrated a continuous network distribution, while the silicon phase displayed a skeleton structure with minimal internal voids. The material exhibited a density of 2.304 g/cm^3^, a maximum thermal conductivity of 111.6 W/m·K, an average thermal expansion coefficient of 8.21 × 10^−6^ K^−1^ from room temperature to 100 °C, and a hardness of 136.8 HBW.(3)The pretreatment of powder using ball milling effectively enhanced the quality of the powder. When utilizing a ball milling duration of 8 h, a ball-to-material ratio of 10:1, and a milling speed of 200 r/min, the most favorable outcomes were obtained. Under these parameters, the material demonstrated superior performance, aligning with the standards for electronic packaging materials.

## Figures and Tables

**Figure 1 materials-16-05763-f001:**
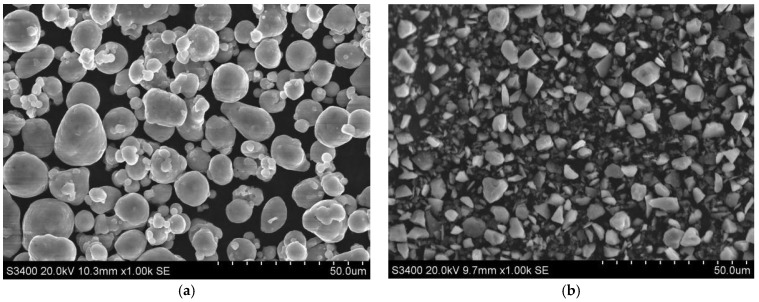
Original powder morphology. (**a**) Al powder, (**b**) Si powder.

**Figure 2 materials-16-05763-f002:**
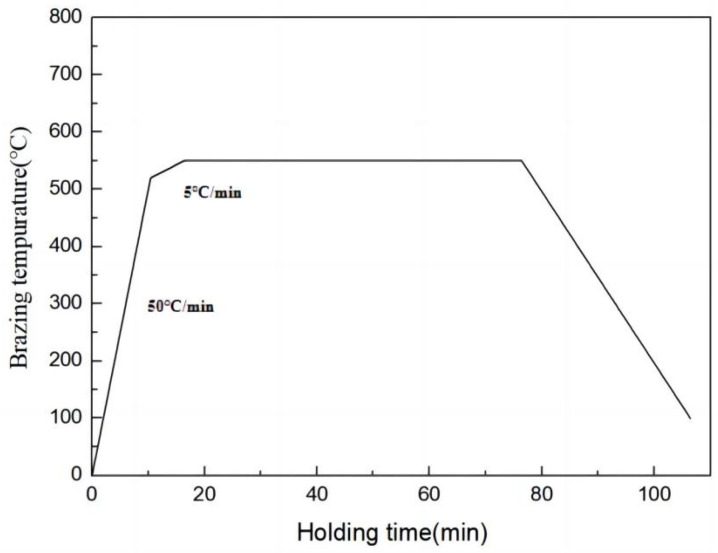
Heating curve of the brazing process.

**Figure 3 materials-16-05763-f003:**
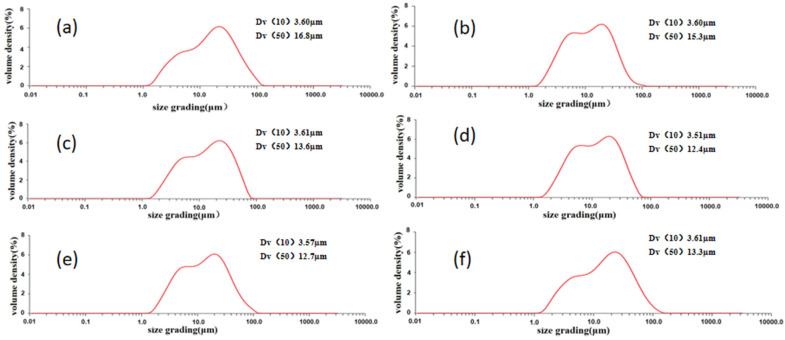
Powder size distribution at different milling times: (**a**) 2 h, (**b**) 4 h, (**c**) 6 h, (**d**) 8 h, (**e**) 10 h, (**f**) 12 h.

**Figure 4 materials-16-05763-f004:**
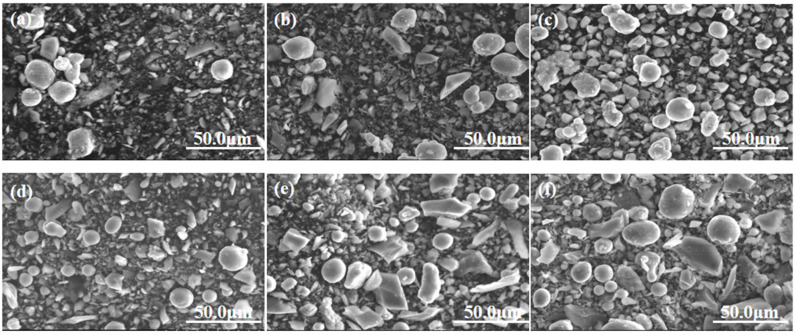
Morphology and distribution of powder at different milling times: (**a**) 2 h, (**b**) 4 h, (**c**) 6 h, (**d**) 8 h, (**e**) 10 h, (**f**) 12 h.

**Figure 5 materials-16-05763-f005:**
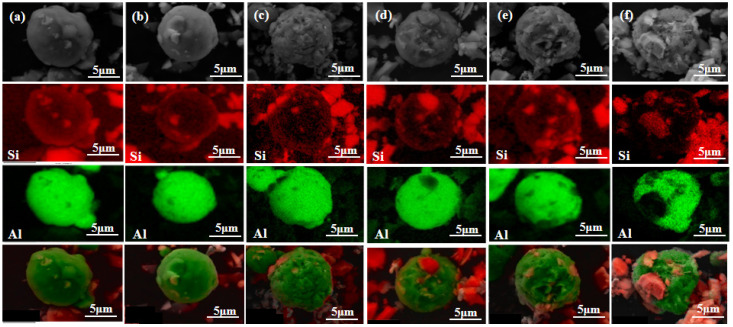
SEM and EDS distribution of some particles enlarged in mixed powder with different milling times: (**a**) 2 h, (**b**) 4 h, (**c**) 6 h, (**d**) 8 h, (**e**) 10 h, (**f**) 12 h.

**Figure 6 materials-16-05763-f006:**
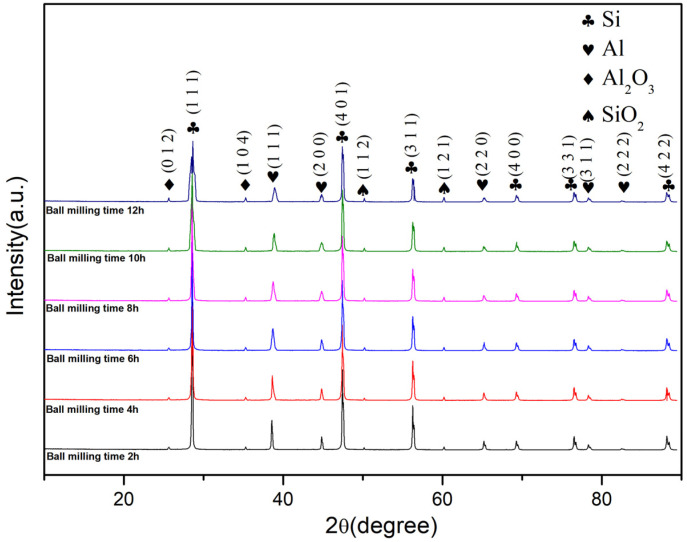
Powder XRD at different milling times.

**Figure 7 materials-16-05763-f007:**
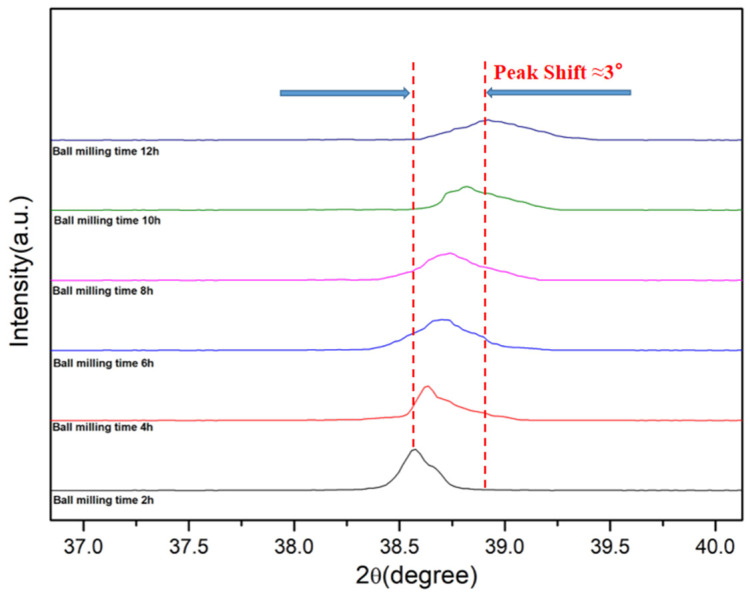
Displacement of the Al (111) reflection peak with milling time.

**Figure 8 materials-16-05763-f008:**
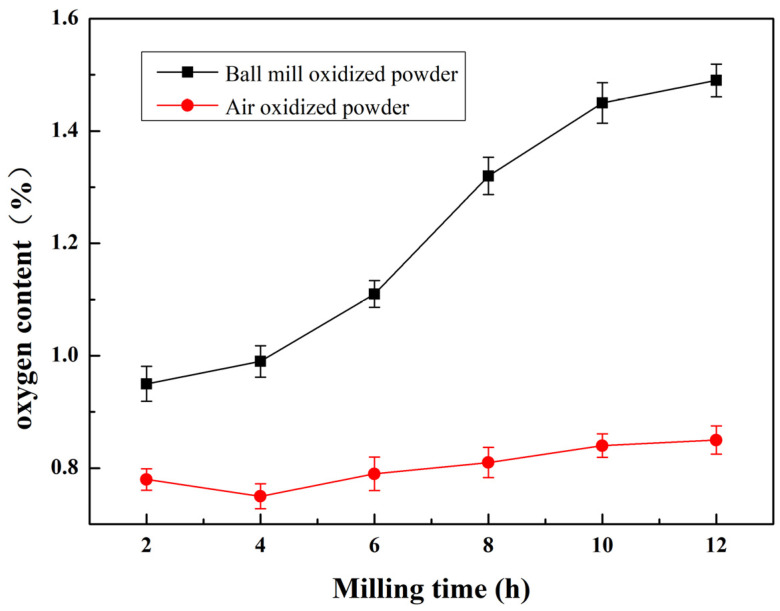
Dependence of oxygen content on the milling time in air and ball milling.

**Figure 9 materials-16-05763-f009:**
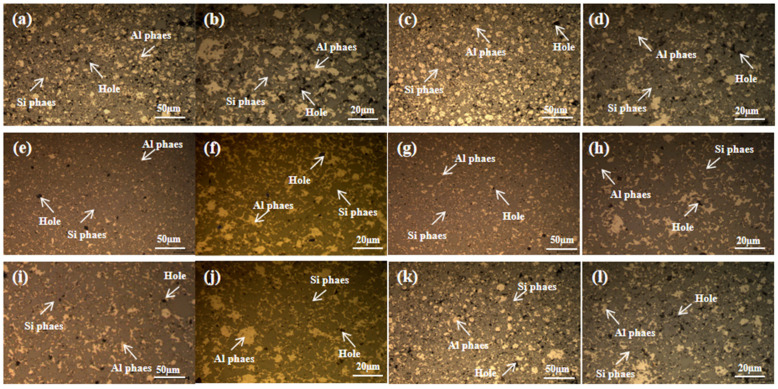
The microstructure of the Al-Si composites was observed at different milling times: (**a**,**b**) 2 h, (**c**,**d**) 4 h, (**e**,**f**) 6 h, (**g**,**h**) 8 h, (**i**,**j**) 10 h, (**k**,**l**) 12 h.

**Figure 10 materials-16-05763-f010:**
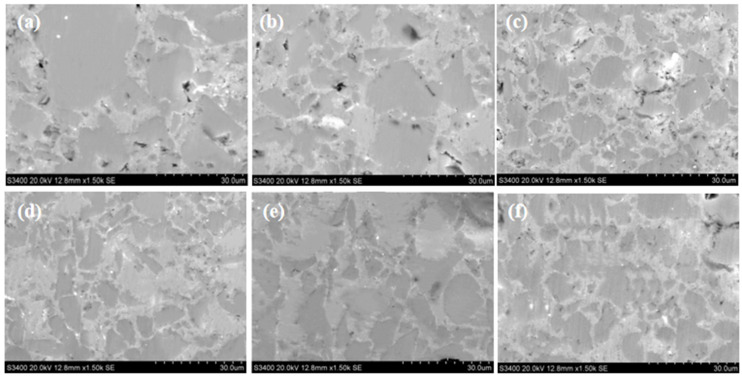
SEM of the Al-Si composites at different milling times: (**a**) 2 h, (**b**) 4 h, (**c**) 6 h, (**d**) 8 h, (**e**) 10 h, (**f**) 12 h.

**Figure 11 materials-16-05763-f011:**
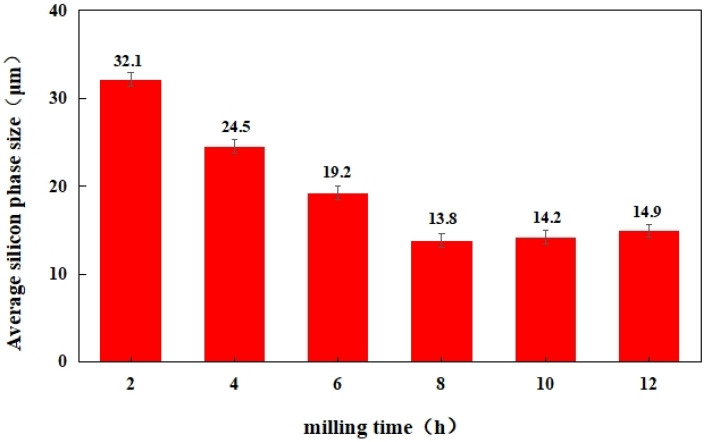
Average silicon phase size.

**Figure 12 materials-16-05763-f012:**
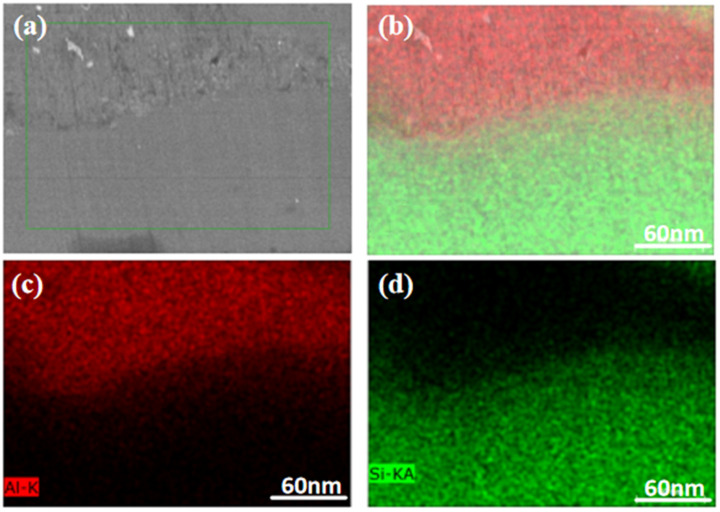
SEM of the Al-Si interface at a ball milling time of 8 h. (**a**) Aluminum-silicon bond interface surface sweep, (**b**) Aluminium and Silicon element, (**c**) Aluminium element, (**d**) Silicon element.

**Figure 13 materials-16-05763-f013:**
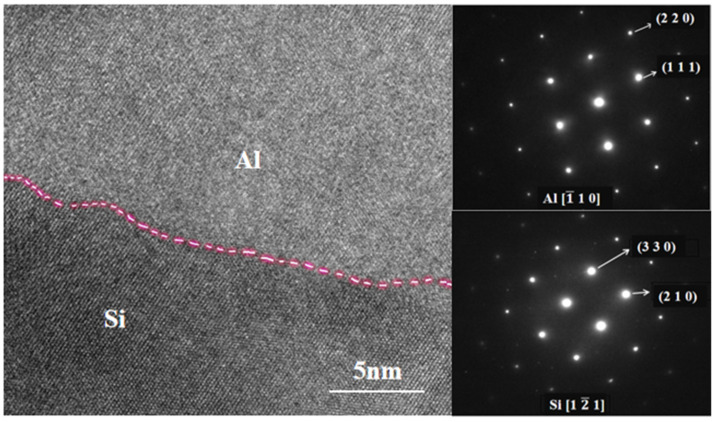
HRTEM diagram and FFT diagram of the Al-Si powder after 8 h of ball milling.

**Figure 14 materials-16-05763-f014:**
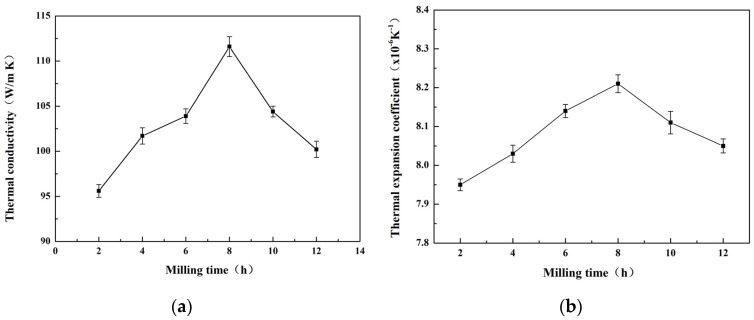
Effect of the milling time on the thermal conductivity and thermal expansion coefficient of the Al-Si composite: (**a**) thermal conductivity, (**b**) thermal expansion coefficient.

**Figure 15 materials-16-05763-f015:**
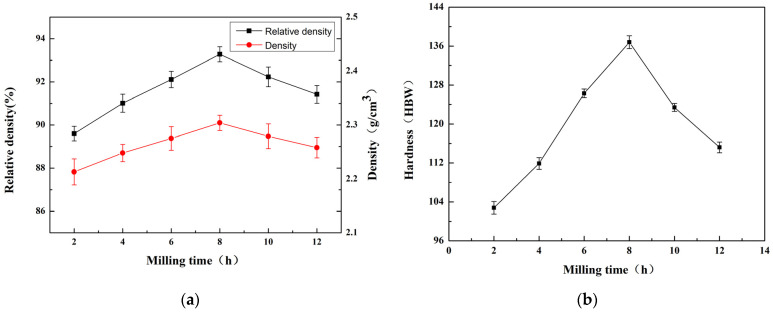
Effect of the milling time on the density, relative density, and Brinell hardness: (**a**) density and relative density, (**b**) hardness.

**Table 1 materials-16-05763-t001:** Main raw materials used in the experiments.

Element	Purity (wt.%)	Median Particle Size (µm)
Al	≥99.5	25
Si	≥99.5	10

## Data Availability

The data used to support the findings of this study are available from the corresponding author upon request.

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
