# Peer review of "Effect of Ball Milling Time on the Microstructure and Properties of High-Silicon–Aluminum Composite"

_materials, 2023, doi:10.3390/ma16175763_

Round 1

Reviewer 1 Report

Dear Authority,

The manuscript entitled ‘Effect of ball milling time on the microstructure, thermal conductivity and hardness of high silicon-aluminum compositeinvestigates the effect of ball milling duration on both physical, mechanical and thermal properties in Al-60Si materials system. Increasing milling duration up to 8h leads particle refinement with monolithic particle size distribution. Further milling process increases the particle size due to cold welding and severe plastic deformation mechanism which is the nature of high energy ball milling process. the composition milled 8h provides homogenous integrity which results better density and hardness values along with improvement on thermal properties.

I think, the paper includes important information and data which will be useful for literature. It could be considered for publication after minor correction according to following comments/recommendations;

1- In this study, the composition you worked on is far away from eutectic point on Al-Si phase diagram. What is the reason you choose this composition rate? Please specify.

2- Please write how to provide Al and Si powder (such as gas atomization) or the company you purchased.

3- Have you used any process control agent in this study, please add that information.

4- If you used any process control agent (PCA), do you think PCA has effect on oxidation level of milled powders? Please discuss accordingly.

5- In XRD data, have you seen any changes on the peak position referring solid solution which is commonly observed with high energy ball milling process? According to Al-Si phase diagram, Al has ability dissolve Si. Probably you have to see changes peak position on 111 reflection of Al as a function of time. Please refer following papers in your manuscript;

https://doi.org/10.1016/j.mtcomm.2021.102202     

6- Reference list can be extended with current issues to provide information about the relationship between milling duration and physical and mechanical properties via high energy ball milling. You can use following manuscript.   https://doi.org/10.1016/j.mtcomm.2023.105868

The manuscript can be published in Materials after these minor corrections.

Best wishes,

Reviewer 2 Report

The manuscript of Kong et al. on the relationship between ball milling time and the properties of Al-Si composites may be advantageous to developers, even though the current version is far from the format and content of a high-level scientific paper. It contains many weaknesses and an inconsistent writing style.

The manuscript needs an in-depth revision, and a native English language check is inevitable before resubmission.

- The keywords include five terms, but three of the five are already in the title. It is an incorrect and unacceptable practice. If authors are not familiar with the purpose of the keywords, they could seek advice from a qualified scholar.

- The whole manuscript, including the title, uses the term high silicon aluminum (composite), but the meaning of "high" is difficult to understand. Is this a geometric parameter or something else? Apart from this inexplicable property, at least three different spellings of this material are in the text (separated by spaces, high-silicon, and high silicon-aluminium). It would be preferable if the authors could go a common ground for this material and stick to a uniform spelling throughout the text.

Al-Si composite has at least three variants (Al40Si(wt%), Al-70%Si, Al60wt%Si)! The previous suggestion also applies to this term.

The authors have failed to give argumentation for studying the 60% Si content. It could be that composites of 40 and 70% Si content are well studied extensively by others, but why the 60% and not 55.5 or 66.7% Si content was the target?

- The Introduction contains a lot of repetition. Please rephrase the whole chapter.

- The Introduction does not include a clear justification of the manuscript. Please clearly inform the reader of the purpose of the manuscript.

- The Material and Methods chapter is incomplete. The authors lack adequate characterization of the instruments used (ball mill, SEM/TEM EM and its parameters, etc., or, e.g., the method for characterizing the particle size distribution), and the masses of the materials used are missing. This chapter provides incomplete information on the reproduction of the experiments.

Authors - in theory - measured the oxygen content of the samples, but they failed to mention the method and the instrumentation.

- Most of the figures are misplaced. The SEM/TEM/particle size distribution figures should be where the authors discuss the microstructure (subchapter 3.2.1).

The caption to Figure 3 says something about particle size distribution. The author seems to have failed to include a figure on particle size distribution calculated from EM images.

- The average size of the silicon phase in Figure 9 is mysterious. How was this parameter measured?

- The abbreviation HRTEM is not defined.

- Journal names are unabbreviated in the Reference section. However, many titles also contain a secret [J] string. What does this mean? Strangely, almost all article titles end with these three characters.

Many repetitions and inconsistent spelling. The entire Introduction needs a reformulation. A native English chemist review is necessary.

Reviewer 3 Report

The manuscript entitled “Effect of ball milling time on the microstructure, thermal conductivity and hardness of high silicon-aluminum composite” presents the development of high Si content Al-Si alloy prepared by ball milling. Although there are similar works on this subject yet some interesting results have been obtained. Some issues that need to be taken care of are as follows.

1.      The originality of this article should be added in introduction section. New alloys discovered such as high entropy alloys should be included. Some of the recent literature can be useful, for instance Powder Metallurgy 64 (3), 192-197 (2021); Journal of Alloys and Compounds 947, 169545 (2023), Archives of Metallurgy and Materials 65 (4) (2020)

2.      The quality of Fig. 2 is poor. Please enhance the clarity.

3.      Why did you choose 12 milling? The silicon content is high. Did you get proper homogenization?

4.      The scale bars in Fig. 3 and 4 should be enhanced. The labels should be made clear.

5.      Please add the respective JCPDS cards in X-ray patterns of Fig. 5.

6.      What is the error in measurement in each data point shown in Fig. 6?

7.      What do the dark and light regions correspond to in Fig. 7?

8.      There are many scratch marks in Fig. 8. Please replace this image.

9.      How did you calculate the size of Si phase from SEM image shown in this work?

10.   Fig. 10 is not acceptable at all. It sounds like an undergrad student’s lab report.

11.   There is no scale bar in Fig. 11. The indexed patterns are also not clear.

12.   Please re-draw Fig. 12 and 13.

13.   Additionally, English should be polished as well. 

Minor editing required.

Round 2

Reviewer 2 Report

The authors significantly improved their manuscript, which is almost suitable for publishing. The authors gave correct feedback to the referee's concerns. 

Only one further correction seems necessary before the appearance: the journal abbreviations need dots. Because this action affects practically all references, a minor revision is advisable.

The language has been improved, and now it is acceptable.

Reviewer 3 Report

The authors have improved the manuscript. 

Minor editing needed. 
